# LevyScore: A Fast Sample-Wise Confidence Score of Pretrained Joint Embedding Model

**Lucas Maes**[1,2*]**, Damien Scieur**[2,3]**, Randall Balestriero**[4]
[1]Mila, [2]Université de Montréal (DIRO)
[3] Samsung SAIL Montreal, [4]Brown University

## Abstract

Foundation models–Deep Networks (DNs) able to solve numerous downstream task without requiring to be retrained–have made enormous strides in the recent years. Thus far, progress has mostly been measured in terms of average performance on mostly curated datasets. Yet, a large number of end-users are concerned with sensitive applications for which an assessment of the foundation model's confidence is required. To that end, we propose **LevyScore**–a simple, fast sample-wise confidence score for any pretrained foundation model using joint-embeddings. LevyScore is theoretically sound as it captured the deviation of an embedding from its pretraining density. Yet, LevyScore *does not require knowledge of the pretraining data nor having access to any downstream dataset*. Instead it is built from a core principle of Joint Embeddings: producing Gaussian embeddings. Our experiments demonstrate that LevyScore provides an effective mechanism for filtering samples according to the foundation model's confidence. Across probes and datasets, it consistently improves the accuracy–coverage tradeoff, achieving state-of-the-art performance. By selectively discarding uncertain predictions, LevyScore offers a simple, principled, and practical tool for deploying foundation models in high-stakes applications.

## 1  Introduction

For decades, machine learning systems took the form of probabilistic models [3, 12]. Whether generative or discriminative, they offered the option to get an estimate of the model's confidence in the current input and prediction [5]. However, that paradigm drifted away leading to today's state-of-the-art systems being entirely trained without offering such scoring mechanisms [8].

For a wide range of applications, such scoring is not necessary, as the model is designed to produce a prediction regardless of its confidence [9]. More advanced pipelines may leverage a mixture of experts to aggregate multiple systems synergistically [10, 4]. However, there exist several applications that require such confidence prediction, as sometimes, not making a prediction is better than making a random one [13, 19].

The common approach today is to take a pretrained foundation model, estimate the density of its embeddings on a dataset, and then rely on this external estimator to measure the model's confidence [6]. However, the outcome is highly sensitive to the choice of estimator, the dataset used for fitting, hyperparameters, *etc* [16]. Additionally, most density estimators incur significant computational costs. For instance, Gaussian Mixture Models require restrictive covariance assumptions to avoid cubic complexity in the dimension, or kernel methods scale quadratically with the size of the training set.

---

[*]Correspondence to `lucas.maes@mila.quebec`

**Contribution.** To address this limitation, we propose **LevyScore**, a training-free method inspired by Lévy's concentration theorem. The approach builds on the fact that joint-embedding foundation models are trained to produce Gaussian embeddings: scoring confidence then reduces to comparing a sample's embedding against the Gaussian distribution it should follow. LevyScore models the embedding space with a Chi-distribution, making its computation depends only on the norm of each embedding. Our experiments show that LevyScore serves as an cheap and effective filtering mechanism. We hope that our very first step in estimating foundation models' confidence will open numerous avenues to easy the deployment of those models in sensitive applications.

## 2 LevyScore: Training-Free Confidence Score for Foundation Models

### 2.1 Norms as a Cheap Sample-Wise Statistic

Most Joint-Embedding Predictive Architectures (JEPA) [14] are trained with two main objectives: (i) *predictive invariance*, ensuring that different "views" of the same input produce similar embeddings, and (ii) *representation diversity*, preventing collapse to trivial solutions [11]. These two principles are typically captured by the following loss:

$$\mathcal{L} \triangleq \sum_{n=1}^{N} \mathbb{E}_{(\boldsymbol{x}_n^{(1)}, \boldsymbol{x}_n^{(2)}) \sim \mathcal{G}(\boldsymbol{x}_n)} \Big[ \mathrm{dist}\big(\mathrm{Pred}(\mathrm{Enc}(\boldsymbol{x}_n^{(1)})), \mathrm{Enc}(\boldsymbol{x}_n^{(2)})\big) \Big] \quad \text{(predictive invariance)}$$
$$+ \mathrm{diversity}\big((\mathrm{Enc}(\boldsymbol{x}_n))_{n \in [N]}\big), \quad \text{(anti-collapse)}, \quad (1)$$

where $\boldsymbol{x}_n^{(1)}, \boldsymbol{x}_n^{(2)}$ are two stochastic views of the same input generated by $\mathcal{G}$, and $\mathrm{dist}$ is a distance function (e.g., $\ell_2$). For image encoders, $\mathcal{G}$ typically consists of two different data augmentations. The diversity (anti-collapse) term plays a crucial role, as it ensures that embeddings do not collapse to a single point, but instead spread out in a manner consistent with a Gaussian distribution [17, 2].

This JEPA training implicitly enforces that the embedding distribution be *spherically invariant*–and in particular Gaussian or Uniform on the surface of the hypersphere as the dimension grows to infinity. In fact, this can be seen both for contrastive learning [18], for non contrastive teacher-student methods [15]. As a direct consequence, the embedding *norms* are expected to follow a Chi-distribution with degrees of freedom equal to the embedding dimension.

This property is central to our work. When attempting to design a *sample-wise* confidence score, one faces a fundamental limitation. Given only a single embedding, it is impossible to assess whether the overall distribution is isotropic or Gaussian. However, the norm of the embedding remains a meaningful and testable statistic.

**Proposition 1.** *Given only one sample, it is not possible to test for spherical invariance. Consequently, the embedding norm is the only statistic that can be exploited for sample-wise confidence estimation.*

This result highlights why norms are the natural building block for a confidence metric. Unlike distributional properties (like the distribution moments) that require many samples to estimate, the norm of an individual embedding already carries information about its compatibility with the expected Gaussian geometry. A sample whose norm lies in the high-density region of the Chi-distribution is consistent with the embedding space geometry. In contrast, samples with unusually small or large norms deviate from this structure, suggesting that the model's predictions could be unreliable.

Motivated by this observation, we define our proposed metric, **LevyScore**, as follows:

$$\mathrm{LevyScore} \triangleq \log \mathrm{PdfChi}_K(\|\boldsymbol{z}\|_2), \quad (2)$$

where $\mathrm{PdfChi}_K$ denotes the probability density of a Chi-distribution with $K$ (the embedding dimension) degrees of freedom. Intuitively, LevyScore assigns higher values to embeddings whose norm lies near the mode of the Chi-distribution, and lower values otherwise. Since it requires no training and depends only on the $\ell_2$ norm of each embedding, LevyScore offers an efficient and theoretically grounded alternative to density estimators that must operate over the full embedding space.

### 2.2 LevyScore in Practice: Norms Reveal Prediction Reliability

To demonstrate the practical utility of LevyScore, we examine how the distributions of embedding norms differ between correct and misclassified samples. In fig. 1, we observe a clear distributional

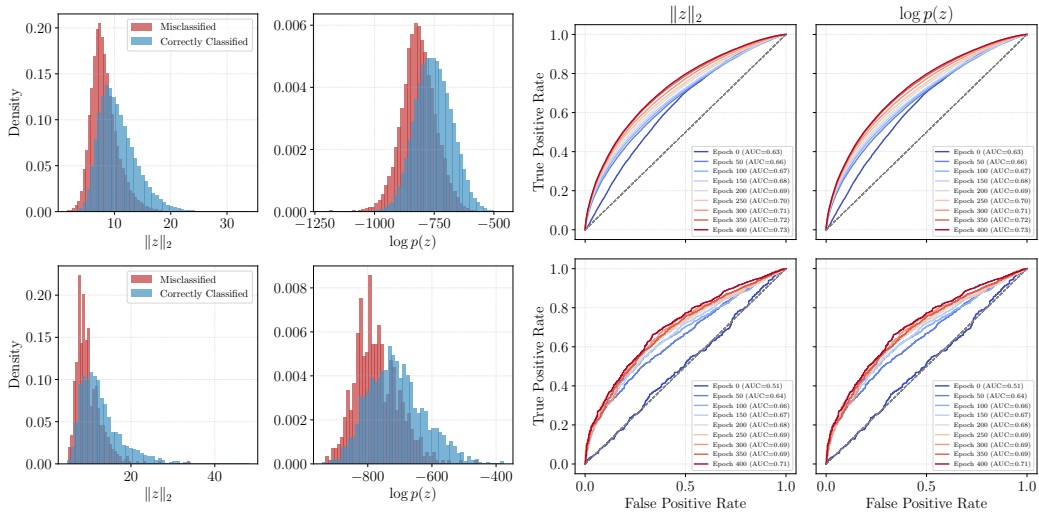

Figure 1: **Left:** Histograms of embedding norms on ImageNet-100 (**Top:** train set, **Bottom:** validation set) using a ResNet-18 backbone trained with SimCLR. Correctly classified samples concentrate around higher LevyScore values, while misclassified samples shift toward lower values, showing that the norm carries discriminative confidence information.**Right:**ROC curves of LevyScore for ImageNet-100 (**Top:** train set, **Bottom:** validation set) with a SimCLR backbone shown across training epochs. As embeddings improve, the separation between correct and misclassified samples increases, leading to steadily higher AUC values.

shift: misclassified samples tend to concentrate in regions with lower LevyScore values, while the correct ones cluster around higher values. This indicates that the embedding norm alone carries information about whether a sample aligns with the model's learned representation.

Further, fig. 1 tracks the evolution of ROC curves throughout training. As the embeddings improve, the separation between the norm distributions of correct and incorrect predictions becomes more pronounced, yielding higher AUC scores. These results suggest that LevyScore can act as an effective rejection criterion: **Top-scoring embeddings reveal the model's most accurate predictions.**

## 2.3   LevyScore: Controlling Accuracy and Filter Tradeoffs

A common procedure for leveraging representations from Joint Embedding Architectures (JEA) is to train a linear classifier (or *linear probe*) on top of the learned embeddings [1]. While these pretrained encoders typically yield more robust representations than those obtained via purely supervised learning [7], in certain applications, the reliability of the probe's predictions is critical (e.g., medical imaging, defense, or other high-stakes settings). In this section, we demonstrate how LevyScore can provide a principled mechanism for improving reliability in such scenarios.

As shown in fig. 2, using LevyScore (or equivalently, the embedding norm) as a rejection threshold on ImageNet-100 predictions enables explicit control over the tradeoff between accuracy and coverage, cf. table 1. Increasing the threshold improves the likelihood that retained predictions are correct, effectively calibrating the model's confidence.

To further validate this claim, we evaluate LevyScore-based filtering across different probe types. In particular, we compare its effect when applied to a $k$-NN probe versus a linear classifier on top of the same encoder embeddings, as shown in fig. 3. In both cases, we observe a consistent phenomenon: filtering samples by LevyScore systematically improves the accuracy of the retained set, confirming its effectiveness as a confidence-aware selection criterion. Additional experiments in the appendix (figs. 4 to 6) further demonstrate that this behavior holds across datasets, probe configurations, and self-supervised learning methods.

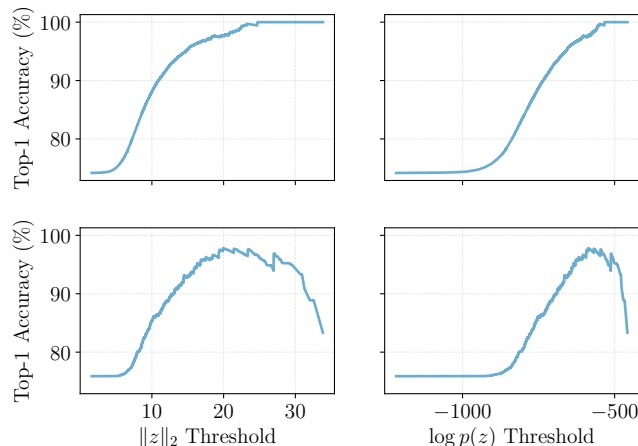

Figure 2: Top-1 accuracy of a linear probe on ImageNet-100 with a SimCLR backbone, **Top:** train; **Bottom** validation. Increasing the LevyScore threshold improves accuracy by rejecting uncertain samples, though overly large thresholds reduce coverage and degrade validation accuracy.

| $\|z\|_2$ | **Train** | **Val** |
|---|---|---|
| 0–10 | 65532 | 1994 |
| 10–20 | 48677 | 2672 |
| 20–30 | 1320 | 316 |
| 30–40 | 10 | 34 |

| $\log p(z)$ | **Train** | **Val** |
|---|---|---|
| -1250 – -1000 | 389 | 0 |
| -1000 – -750 | 68808 | 2128 |
| -750 – -500 | 46315 | 2830 |
| -500 – -250 | 27 | 42 |

Table 1: Samples per bin for ImNet-100. **Left:** $\|z\|_2$ bins (size = 10). **Right:** $\log p(z)$ bins (size = 250).

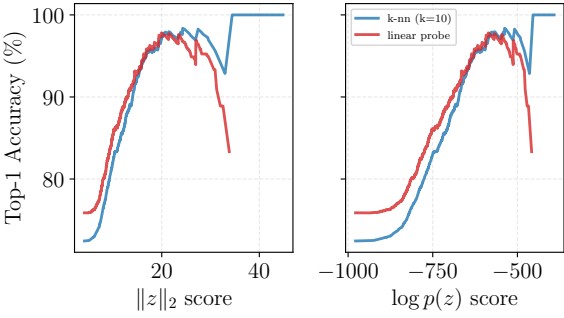

Figure 3: Comparison of $k$-NN (k=10) and linear probes on ImageNet-100 validation embeddings from a SimCLR-trained ResNet-18. Both probes show similar accuracy–coverage improvements when filtering with LevyScore, indicating robustness across probe types.

## 3   Conclusion

We introduced **LevyScore**, a cheap, fast, and principled sample-wise confidence score for pretrained joint-embedding models. By leveraging the assumed Gaussian geometry achieved by joint-embedding training, LevyScore reduces confidence estimation to a simple function of the embedding norm. This provides substantial advantages over traditional density estimators: it is training-free, computationally efficient, and does not require access to downstream data. Our experiments across ImageNet-100 and CIFAR-10 demonstrate that LevyScore effectively separates correctly classified from misclassified samples, tracks improvements in embedding quality during training, and systematically enhances accuracy–coverage tradeoffs across both linear and non-parametric probes. These findings highlight LevyScore as a lightweight yet robust criterion for filtering uncertain predictions.

**Limitations and Future Work.**   While promising, LevyScore has so far been evaluated primarily on vision encoders trained with SimCLR and at relatively modest scales. Future work will extend our study to larger-scale foundation models, multimodal encoders, and out-of-distribution detection settings. Another important direction is exploring how LevyScore can be integrated into training or inference pipelines to actively improve calibration and reliability in high-stakes applications.

## Acknowledgments

LM and DS thank Prof. Simon Lacoste-Julien for his financial support, which funded LM during this project.

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

# A   Additional Experiments

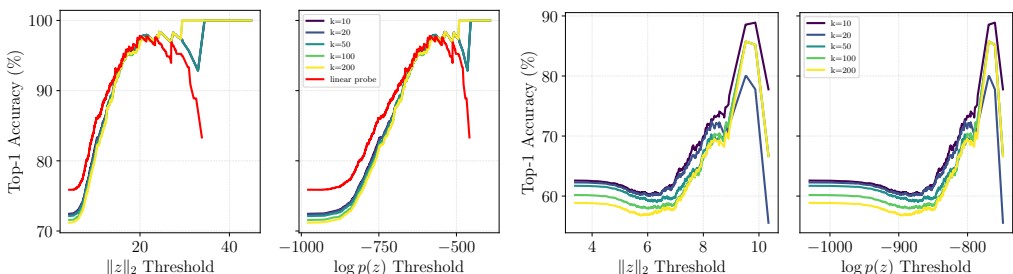

Figure 4: Effect of neighborhood size $k$ in $k$-NN probes on validation embeddings using a SimCLR-pretrained backbone. **Left:** ImageNet-100; **Right:** CIFAR-10. Smaller $k$ values (e.g., 10) yield higher peak accuracy in both datasets. Top-1 accuracy is shown versus the rejection threshold for embedding norm ($\|z\|_2$) and log-density ($\log p(z)$). Across both datasets, LevyScore filtering remains effective for different $k$, illustrating the robustness of norm-based confidence metrics.

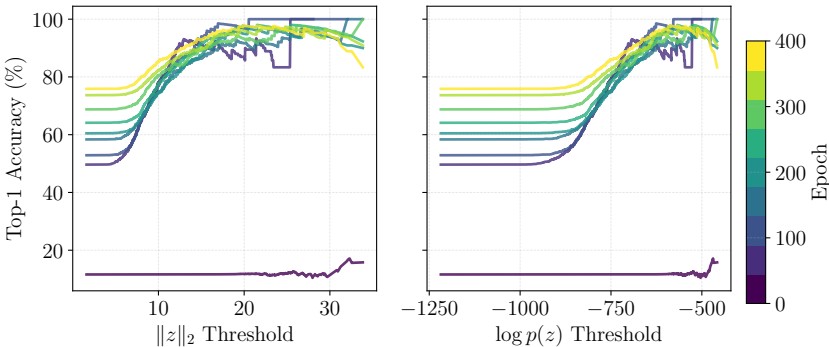

Figure 5: Top-1 accuracy on ImageNet-100 validation set (SimCLR backbone) as a function of LevyScore threshold across training epochs. As training progresses, embeddings improve and higher thresholds lead to more reliable predictions, yielding better accuracy–coverage tradeoffs. Extremely large thresholds eventually reject too many samples, reducing coverage and lowering accuracy.

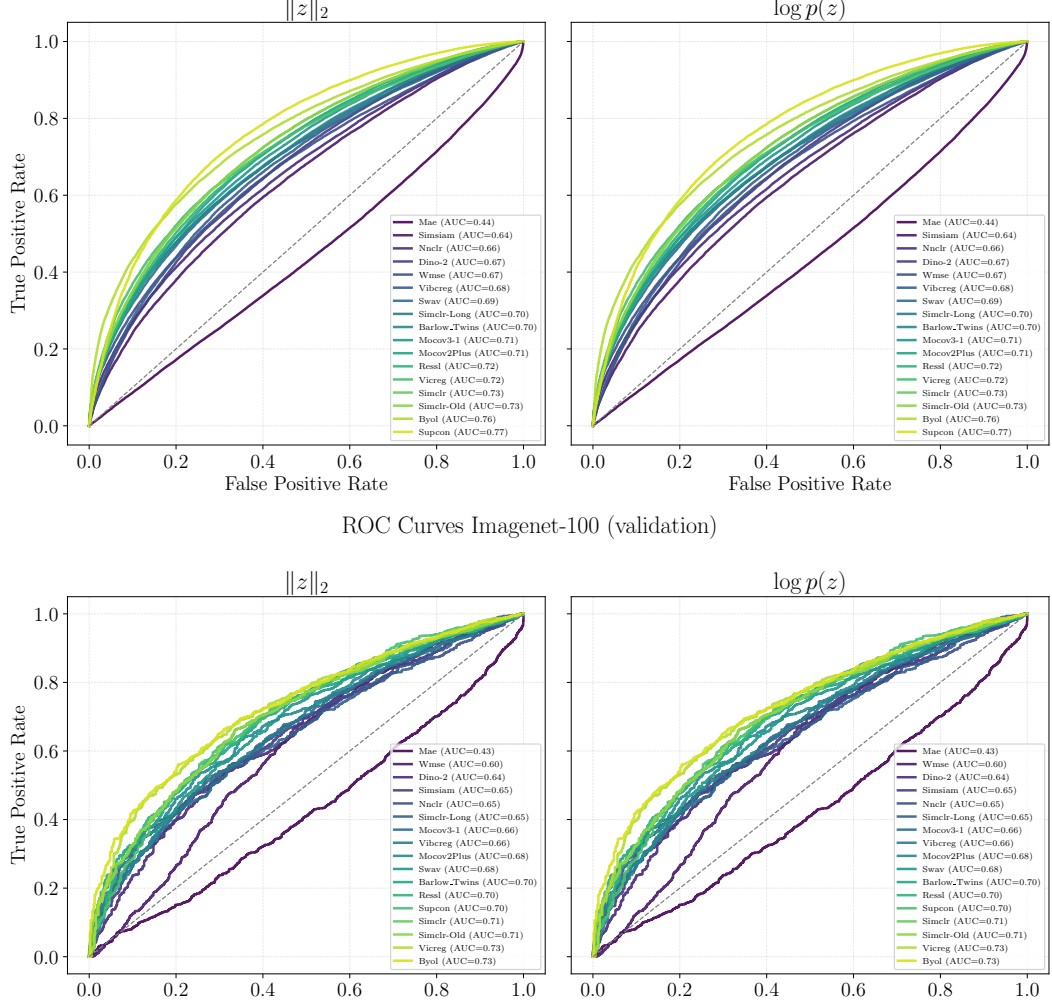

Figure 6: ROC curves of LevyScore on ImageNet-100 (**Top:** taining, **Bottom:** validation set) across multiple self-supervised learning methods. Stronger SSL methods (e.g., BYOL, VICReg) yield better separation, confirming that LevyScore tracks embedding quality across approaches. The only exception is the non-JEA method Masked Auto-Encoder (MAE), which does not benefit from LevyScore, likely due to its different embedding geometry.

