# OpenReview forum: "LevyScore: A Fast Sample-Wise Confidence Score of Pretrained Joint Embedding Model"
_NeurIPS.cc/2025/Workshop/UniReps — UniReps2025_

### Official Review · Reviewer_6Dro · 2025-09-05
**LevyScore: A Fast Sample-Wise Confidence Score of Pretrained Joint Embedding Model**

**Confidence:** 5

**Review:**

The abstract remains underdeveloped, as it does not encapsulate the full scope of the study. A well-crafted abstract should succinctly summarize the research objectives, methodology, key findings, and conclusions. Notably, the results section is absent, which is essential for conveying the impact and relevance of the work.
Nonetheless, the overall assessment reflects a commendable piece of research. Upon addressing the highlighted revisions, the manuscript is considered suitable for publication.

**Score:**

5

**Topic Fit:**

3

---

### Official Review · Reviewer_a731 · 2025-09-15
**Review: LevyScore**

**Confidence:** 4

**Review:**

## Summary
The paper proposes LevyScore, a training-free per-sample confidence score for joint-embedding encoders by evaluating the $\chi$-distribution log-pdf at the embedding norm.

---

## Strengths
- Well written and easy to follow.
- Simple, post-hoc and fast.
- Clear geometric intuition.
- Separation improves with training and appears across multiple SSL variants.

---

## Weaknesses
- SOTA claims should be supported by head-to-head comparisons with confidence baselines (e.g., logit margin/entropy, energy).
- Gaussianity / spherical invariance is central to the method but lacks empirical checks. Empirical evidence on low-dimensional space would strengthen the claim.
- "Norm is the only statistic" can be true in the strict, single-embedding setting. However, as mentioned in the paper, many practical pipelines come with a linear probe, yielding additional confidence signals.
- Reported AUROCs suggest moderate separation. Without seeing any head to head comparisons it is hard to argue how good the method performs.
- The method implicitly assumes that the latent center is the origin. If this is not the case, the model is a non-central $\chi$ distribution, which requires re-centering on an estimated mean, hurting the claim of a single-sample score.

---

## Comments / Questions
1. JEPA terminology: In my opinion Equation 1 is a general SSL setting that fits better to SimCLR or CLIP rather than JEPA.
2. Strong linear probe performance from powerful SSL backbones implies multi-modality in the embedding space. Does this contradict with the $\chi$ distribution claim?

**Score:**

2

**Topic Fit:**

2

---

### Official Review · Reviewer_H6L1 · 2025-09-17
**Interesting idea, still very preliminary**

**Confidence:** 5

**Review:**

This paper introduces an interesting embedding confidence score metric that considers the norm of the embedding compared against statistics of the overall population of embeddings. Overall it's clear that the authors are trying to make a point here: embedding norms are free to calculate, but trained embedding confidence is not, and is frequently unreliable. That said, the way that the point is made is not strong.

The authors use a classification task's precision as a discriminator for demonstrating distributional differences between the embedding norms in Figure 1, but this is not prescriptive: without labels, one doesn't know whether a particular embedding belongs to the correct or incorrect classification groups. If one were to use purely the norm itself and arbitrarily threshold, they would be eliminating a large portion of their classifications. Finally, frequently embeddings are fine-tuned for classification tasks, but such details aren't explained in the context of Figure 1.

The accuracy and filter trade-offs section is highly anecdotal without significant bearing out. The idea could use more comprehensive evidence, not just a deep dive into some of the post-training metrics for image classification as stratified by the authors' chosen binning. It's not that these results are incorrect or unhelpful, it's that they are too narrow to draw broader conclusions from.

Suggestions for improvement would be to expand past single shot metrics on a particular model backbone and linear probe task, and provide a comparison of other techniques for confidence and demonstrate that this approach works better than they do across a range of problems, then a deep dive into how we should think to apply this method. That said, the point of this paper is clear and the topic will likely spur interesting discussion among participants.

**Score:**

3

**Topic Fit:**

3

---

### Official Review · Reviewer_eW1t · 2025-09-18

**Confidence:** 4

**Review:**

## Summary

This paper proposes LevyScore, a training free, sampl wise score for JEPA models. The core idea is that embeddings from contrastive/self-supervised training are under pressure to be approximately Gaussian, so their norms follow a Chi distribution. With that in mind, LevyScore is defined as the log-likelihood of an embedding’s norm under the Chi distribution with degrees of freedom equal to the embedding dimension. Samples with “typical” norms receive high scores, while atypical ones get low scores. Experiments show that LevyScore separates correctly classified from misclassified samples on ImageNet-100 and CIFAR-10, tracks embedding quality across training epochs, and improves the accuracy coverage tradeoff when used as a filtering criterion. It also generalizes across probes (linear classifiers, k-NN) and across self-supervised methods (SimCLR, BYOL, VICReg, etc.), with the exception of MAE, which has different embedding geometry.

## Strengths

* Clarity: The paper is clearly written and well motivated.
* Simplicity: LevyScore requires only the embedding norm.
* Contribution: Provides a fast, principled way to filter predictions from joint embedding models.
* Relevance: Confidence estimation is important for in high stakes settings.

## Weaknesses

* Framing: The paper frames LevyScore as a confidence score, but it is better understood as a typicality score that correlates with prediction reliability.
* Calibration: No evaluation against standard confidence calibration metrics (e.g., ECE, reliability diagrams). Baselines are mostly other SSL methods with LevyScore applied, but not against alternative confidence scoring approaches (e.g., Mahalanobis distance, deep ensembles).
* Scope: Experiments are limited to relatively small vision encoders (ResNet-18, ImageNet-100, CIFAR-10).

## Suggestions (forward-looking)

* Clarify framing: present LevyScore as a norm-based typicality measure that correlates with confidence, unless calibration experiments confirm it as a true confidence score.
* Add calibration experiments (ECE, reliability diagrams) to test whether LevyScore is aligned with probabilistic confidence.
* Compare against alternative confidence scoring baselines beyond density estimators.
* Extend experiments to larger and multimodal foundation models to test generality

**Score:**

4

**Topic Fit:**

2